# Molecular Basis of Transglutaminase-2 and Muscarinic Cholinergic Receptors in Experimental Myopia: A Target for Myopia Treatment

**DOI:** 10.3390/biom13071045

**Published:** 2023-06-27

**Authors:** Veluchamy Amutha Barathi, Candice E. H. Ho, Louis Tong

**Affiliations:** 1Translational Preclinical Model Platform, Singapore Eye Research Institute, 20 College Road, Singapore 169856, Singapore; candice.ho.e.h@seri.com.sg; 2Department of Ophthalmology, Yong Loo Lin School of Medicine, National University of Singapore, National University Hospital System, 10 Medical Dr, Singapore 117597, Singapore; 3Eye-Academic Clinical Program, DUKE-National University of Singapore Gr Medical School, 8 College Road, Singapore 169857, Singapore; louis.tong.h.t@singhealth.com.sg; 4Corneal and External Eye Disease, Singapore National Eye Centre, 11 Third Hospital Avenue, Singapore 168751, Singapore; 5Ocular Surface Research Group, Singapore Eye Research Institute, 20 College Road, Singapore 169856, Singapore

**Keywords:** myopia, transglutaminase-2 (TGM-2), muscarinic cholinergic receptors (mAChRs), scleral growth, TGase inhibitors, mAChRs antagonists, intervention

## Abstract

Myopia, a prevalent refractive error disorder worldwide, is characterized by the elongation of the eye, leading to visual abnormalities. Understanding the genetic factors involved in myopia is crucial for developing therapeutic and preventive measures. Unfortunately, only a limited number of genes with well-defined functionality have been associated with myopia. In this study, we found that the homozygous TGM2-deleted gene in mice protected against the development of myopia by slowing down the elongation of the eye. The effectiveness of gene knockdown was confirmed by achieving a 60 percent reduction in TGM-2 transcript levels through the use of TGM-2-specific small interfering RNA (siRNA) in human scleral fibroblasts (SFs). Furthermore, treating normal mouse SFs with various transglutaminase inhibitors led to the down-regulation of TGM-2 expression, with the most significant reduction observed with specific TGM-2 inhibitors. Additionally, the study found that the pharmacological blockade of muscarinic receptors also slowed the progression of myopia in mice, and this effect was accompanied by a decrease in TGM-2 enzyme expression. Specifically, mice with homozygous mAChR5, mAChR1, and/or mAChR4 and knockout mice exhibited higher levels of TGM-2 mRNA compared to mice with homozygous mAChR2 and three knockout mice (fold changes of 5.8, 2.9, 2.4, −2.2, and −4.7, respectively; *p* < 0.05). These findings strongly suggest that both TGM-2 and muscarinic receptors play central roles in the development of myopia, and blocking these factors could potentially be useful in interfering with the progression of this condition. In conclusion, targeting TGM-2 may have a beneficial effect regarding myopia, and this may also be at least partially be the mechanism of anti-muscarinic drugs in myopia. Further studies should investigate the interaction between TGM-2 and muscarinic receptors, as well as the changes in other extracellular matrix genes associated with growth during the development of myopia.

## 1. Introduction

Myopia is a common refractive disorder affecting a large portion of the global population, particularly in Asian countries like Singapore, where prevalence rates of up to 80% have been reported for those up to the age of 18 [1]. Although myopia can be corrected using refractive methods like spectacles, lenses, or corneal surgery, the underlying pathology increases the risk of serious eye conditions like atrophy of retinal pigment epithelium, glaucoma, and retinal detachment [2]. These complications are not treatable via refractive correction, making high myopia a leading cause of blindness in countries like Japan, Taiwan, and Singapore [3].

Myopia is a condition where the normal process of emmetropization, which occurs within the eye, fails to function properly. Emmetropization involves the detection of defocus by certain cells in the outer retina, such as the amacrine and bipolar cells. This information is then transmitted across the retinal pigment epithelium and choroid through a signal or signals. The scleral matrix, which is the structure surrounding the eye, is also modified, likely through the regulation of proteoglycan synthesis [4]. Our understanding of myopia has been improved through clinical experience, longitudinal studies, epidemiological data, and various animal experiments. However, distinguishing between environmental and genetic influences, especially those related to gradual developmental changes, can complicate the matter.

To study emmetropization and the development of myopia, researchers have utilized different animal models, including chicks, rabbits, tree shrews, guinea pigs, mice, and macaques [5]. These models have been instrumental in characterizing the optical parameters and investigating the mechanisms behind myopia. Hypotheses regarding myopic development have been based on studies examining chicks’ eyes, although the absence of a fovea (a central area of high visual acuity) and retinal blood supply in chicks raises questions about the applicability of these findings to other species. Even closely related primate species can exhibit varying responses to conditions that induce form deprivation, suggesting differences in the mechanisms controlling eye growth. For example, monocular occlusion in the rhesus macaque leads to myopia when the ciliary muscle is paralyzed or the optic nerve is cut, but this does not occur in the stump-tailed macaque. These differences imply that excessive accommodation plays a role in myopia development in the stump-tailed macaque but not in the rhesus macaque.

Considering the variability among animal models, it is crucial to continually evaluate the relevance of each model to the human condition in ongoing myopia research. One promising avenue for future investigation lies in studying changes in gene expression patterns during emmetropization and myopic progression, both within and across species. Identifying the deficient steps in the regulatory pathway would be a significant breakthrough, given the substantial impact of myopia.

The molecular basis of myopia is not yet fully understood, although research has suggested that the axial elongation of the eye, mediated by alterations in the connective tissues of the fibrous sclera coat, is a contributing factor [6,7]. While various polymorphisms in scleral extracellular matrix (ECM) molecules have been linked to myopia, no single candidate gene fully accounts for childhood developmental myopia, the most common type of human myopia [8].

Various factors influence the remodeling of the scleral connective tissue and axial elongation. Genetic predisposition, environmental factors, and visual behaviors such as prolonged work and reduced outdoor activities have been implicated in the development of myopia. The combination of these factors disrupts the delicate balance between scleral growth and the remodeling processes, leading to the elongation of the eye [9].

Understanding the intricate interplay between connective tissue remodeling in the sclera and axial elongation is vital for developing effective strategies to prevent and manage myopia [10]. Recent advancements in imaging techniques, molecular biology, and biomechanical studies have provided valuable insights into these underlying mechanisms. This knowledge opens new avenues for therapeutic interventions and the development of novel treatment approaches targeting the structural and biochemical aspects of the sclera.

Recent studies [11,12,13] have shown that transglutaminase 2 (TGM-2), a multi-functional enzyme known for its role in wound healing and ECM remodeling, is expressed in mouse and human scleral fibroblasts (SF) and that it plays a central role in scleral growth and axial elongation. Fibronectin and collagen in the ECM can trigger outside-in signals in a TGM-2 dependent fashion. TGM-2 may also be involved in the remodeling of ECM via the covalent cross-linking of ECM molecules, affecting the stabilization and degradation of these molecules, or in regulating the motility, adhesion, and survival of the fibroblasts that produce ECM [14].

Experimental myopia associated with the axial elongation of the eye in relation to increased vitreous chamber has been validated, and our previous study and recent studies have suggested that myopiagenesis through the remodeling of the scleral ECM and controlling scleral growth and axial elongation [13,15,16]. This study aims to investigate whether the role of TGM-2 is critical in a mouse model of myopia, the growth-related changes in TGM-2, and its interaction with muscarinic cholinergic receptors (mAChRs) during myopia development; to determine the effect of mAChRs agents on SF derived from TGM-2 knockout (KO) mice; and to identify novel targets for myopia treatment. By examining our latest research findings, we aim to shed light on the complex processes involved in the development and progression of myopia. Ultimately, this knowledge can pave the way for more targeted and effective interventions to address the growing global burden of myopia.

## 2. Materials and Methods

### 2.1. Animal Use and Ethical Approval

Heterozygous mAchR1-5 KO mice (courtesy of Jurgen Wess) and Heterozygous TGM-2 KO mice (courtesy of Robert M Graham and I.Sirri) were backcrossed twelve generations to C57BL/6J wild-type (WT) mice to achieve genomic homogeneity of 99.95% [17], then crossbred in the animal holding unit of Singhealth Experimental Medicine Center and genotyped. We used 6 pairs of breeding mice in each strain to generate more knockout animals. These animals were systemically healthy and generate offspring readily. A background breeding pair was available for backcrossing in every 3–4 months to maintain the homozygosity. Both male and female mice (at 10 days of age) were utilized to induce experimental myopia, and the experiments ended once they reached 52 days of age. Naive control animals were housed in groups of 5, while experimental animals were housed individually after turning 28 days old at 25 °C and subjected to a light–dark cycle (ratio—12:12 h of light/darkness), with mouse pellets and water being available ad libitum. Approval was obtained from the SingHealth Institutional Animal Care and Use of Committee (IACUC; AAALAC accredited), and all aspects of the study complied with the Association for Research in Vision and Ophthalmology (ARVO) statement for animal experimentation.

### 2.2. Murine Myopia Model

Experimental myopia was induced for 6 weeks in TGM-2 KO mice and WT mice (60 mice from each strain for each treatment) by attaching −15 diopter spectacle lens or diffusers over the right eyes. Left eyes were used as controls. A −15 diopter spectacle lens (PMMA Contact Lens in Grey Tint, 8.5 mm diameter, 8 mm base curve, refractive Index: 1.43, axial thickness: 0.5 mm) was placed over the right eye on postnatal day 10 by gluing to an annulus of Velcro, and then attaching to a matching piece of Velcro that had been previously sutured to the skin around the eye. The spectacle lenses were cleaned daily in dim light, and the left eye was left uncovered, serving as the control. In parallel, other groups of 10 mice in each strain had a plano-powered lens attached, using materials like that of the minus lens in the right eyes. All optical interventions were removed on postnatal day 52.

### 2.3. Eye Biometry and Refractive Error Assessment

Each eye was refracted at the beginning of each week to measure the refractive error by using an automated infrared photorefractor. By using an Optical Low Coherence Interferometer (OLCI, AC Master, Carl-Zeiss, Oberkochen, Germany), the axial length of the eye was measured in vivo as previously described [18]. This method has been demonstrated to provide improved resolution and reproducibility and allowed for the serial monitoring of biometry of the eyeball in various stages of myopic induction.

### 2.4. Human Tissues

Human scleral tissues (*n* = 6 eyes) harvested at autopsy within 24 h from normal cadaver eyes (age range—45–80 years; no history of refractive error record) were provided by the Singapore Eye Bank. The protocol was approved by the Institutional Review Board of the Singapore Eye Research Institute. All study procedures were performed as part of standard clinical care and complied with the tenets of the Declaration of Helsinki regarding human research, and as all procedures performed were essential for standard clinical care of these patients, written informed consent was not required, but consent was obtained by assent. The patient’s next of kin were aware of the privacy policy of the hospital that states that information released for publication would not include patient identifiers.

### 2.5. Scleral Fibroblast Cell Culture

The scleral tissues were harvested from human cadavers (*n* = 6 eyes), TGM-2 KO mice, and WT mice (*n* = 80; 20 eyes/batch), and fibroblasts were cultured from explants. At near confluency, the cells were trypsinized and seeded in serum-containing DMEM-based medium. After 4 h, freshly prepared muscarinic agents, such as atropine, carbachol, AFDX-116, himbacine, darifenacin (Sigma, St. Louis, MO, USA), were added at concentrations of 0.01, 0.1, 1, 5 and 10 µM. We aimed to investigate the effects of inhibiting TGM-2 in cultured scleral fibroblast cells using the following TGM-2 inhibitors: monodansylcadaverine (MDC) with an IC_50_ value of 9 µM (concentration: 5, 10, 25 µM); KCC009 (a dihydroisoxazole inhibitor), with an IC_50_ value of 1 µM (concentration: 10, 25, 50 µM); Z006 (Z-DON-Val-Pro-Leu-OMe), a peptide-based site-specific inhibitor, with IC_50_ approx. 0.1 µM to TGM-2 (concentration: 40, 55, 70 µM); and D003 (1,3-Dimethyl-2-[(2-oxo-propyl)thio]imidazolium chloride), which is an active site-directed inhibitor of transglutaminase, with IC_50_ of about 0.8 µM for TGM-2 at concentrations of 1, 10, 25 µM (Zedira GmbH, Darmstadt, Germany). Sterile incubation was carried out in 5% CO_2_ and at 37 °C. After 24 h, medium was removed, and fresh medium was added with each drug. This was repeated daily to avoid oxidation. At day 5, total cellular RNA and proteins were collected for RT-qPCR and protein assays (western blot or transamidase activity), respectively.

### 2.6. Sample Preparation for Protein Assay

The cells (*n* = 3 sets; 1 × 10^6^ cells/set) were lysed via sonification in 1x RIPA lysis buffer (Santa Cruz Biotechnology, Santa Cruz, CA, USA) with 10 µL PMSF solution, 10 µL sodium orthovanadate solution, and 20 µL protease inhibitor cocktail solution. After centrifugation at 20,000× *g* at 4 °C for 20 min, supernatants were collected. The protein content in the supernatants was measured by using the DC Protein Assay kit (Bio-Rad, Berkeley, CA, USA) and following the manufacturer’s instructions. Samples were stored at −80 °C until assayed.

### 2.7. Immunohistochemistry (IHC)

The whole mouse eye (2 months old, *n* = 6 from each group) was embedded in OCT (frozen tissue matrix) compound at −20 °C for 1 h. Prepared tissue blocks were sectioned with cryostat at the thickness of 5 microns and collected on clean polysine™ glass slides. Sections were fixed with 4% paraformaldehyde for 10 min. After washing (3× with 1× PBS) for 5 min, 4% goat serum diluted with 1× PBS was added as a blocking buffer. The slides were then covered and incubated for 1 h at room temperature (RT) in a humid chamber. After rinsing with 1× PBS, a specific primary antibody for TGM-2, mAChR1-5 (polyclonal raised in rabbit, Abcam, Cambridge, UK) diluted (1:100) with 2% goat serum was added and incubated further at 4 °C in a humid chamber overnight. After washing 3× with 1× PBS for 10 min, fluorescein-labelled goat anti-rabbit secondary antibody (1:200, Chemicon International, Temecula, CA, USA) was applied and incubated for 90 min at RT. After washing and air-drying, slides were mounted with anti-fade medium containing DAPI (4, 6-diamidino-2-phenylindole; Vectashield; Vector Laboratories, Burlingame, CA, USA) to visualize the cell nuclei. Sections were incubated with 2% goat serum, with omitted primary antibodies used as controls. A fluorescence microscope (Axioplan2; Carl Zeiss Meditec, GmbH, Oberkochen, Germany) was used to capture images. Experiments were repeated in triplicates from 3 different samples from each group.

### 2.8. Western Blotting

Proteins in the supernatant were separated via SDS-PAGE, transferred to nitrocellulose membranes, blocked in 5% BSA in TBST (10 mM Tris-HCl [pH 8.0], 150 mM NaCl, and 0.0.05% Tween-20) for 2 h at RT, and incubated with the same anti-TGM-2 antibody described above at a dilution of 1:1000 for 1 h at RT, and anti β-tubulin antibodies were used as loading controls. The membranes were washed 3 times in TBST and incubated with HRP-conjugated secondary antibodies (Chemicon International) at a dilution of 1:2500 for 1 h at RT. Immunoreactive bands were visualized using the enhanced chemiluminescence method (GE Healthcare, Buckinghamshire, UK). The membrane was wrapped in plastic and placed against an X-ray film for an appropriate length of time (30 s–5 min).

### 2.9. Quantitative Assay for Cell Surface TGM-2 Activity

This assay assesses the cross-linking enzymatic activity on peptides (TG-Covtest, Covalab, Cambridge, UK). This is a quantitative colorimetric transamidase assay. TGM activity associated with the extracellular surface was measured via the incorporation of fibronectin into biotinylated cadaverine. For this assay, 2 × 10^5^ cells/mL were plated into 96-well plates precoated with plasma fibronectin in 100 μL complete DMEM without serum but containing 0.1 mM biotinylated cadaverine. As a negative control, 96-well plates coated with fibronectin were incubated with 100 μL serum-free DMEM containing 0.1 mM biotinylated cadaverine in the absence of cells. Scleral fibroblasts were incubated for 1 h at 37 °C and then washed twice with PBS (pH 7.4), containing 3 mM EDTA, to stop the reaction. A detergent solution (100 μL) consisting of 0.1% (*w*/*v*) deoxycholate in PBS (pH 7.4) and 3 mM EDTA was then added to each well followed by incubation with gentle shaking for 20 min. The supernatant was discarded, and the remaining fibronectin layer was washed three times with Tris-HCl (pH 7.4). Cells were blocked with 3% (*w*/*v*) BSA in Tris-HCl buffer for 30 min at 37 °C and washed with buffer. The incorporated biotinylated cadaverine was revealed using a 1:5000 dilution of peroxidase conjugate (Extravidin; Sigma-Aldrich), which was incubated for 1 h at 37 °C with TMB as a substrate. Color development was stopped by adding 50 μL stop solution to each well. The resultant color was then read with an ELISA plate reader at 450 nm.

### 2.10. Scleral Fibroblast Cell Proliferation Assay

Roche Xcelligence system RTCA SP (Roche Applied Science, Basel, Switzerland) was used for the monitoring of SF cell proliferation in real-time. Cells were seeded in microtiter plates containing microelectronic sensors (96× E-Plate). The interaction of cells with the electronic biosensors generated a cell–electrode impedance response, expressed as cell index, allowing cell numbers to be detected.

Cells that had previously been transfected were trypsinised and counted. A total of 5000 TGM-2 KO and WT SF cells were seeded in 100 μL of media in duplicates in a 96× E-Plate; proliferation was monitored in real-time. Cell–sensor impedance was measured every 5 min for the first 2 h and then every 15 min for the next 7 h.

### 2.11. RNA Interference Assay

Human scleral fibroblasts (HSF) of Passage 4 were used. Additionally, 3.5 × 10^4^ HSF cells were plated in 12-well plates and maintained in DMEM supplemented with 10 percent FBS and 1% penicillin and 1% streptomycin at 37 °C (5% CO_2_) prior to transfection. To block the function of TGM-2 in HSF cells, we used small interfering (si)RNA molecules targeted at TGM-2 mRNA. All siRNA used were purchased from Qiagen (homosapien transglutaminase 2 (TGM2), transcript variant 1, mRNA NCBI, Reference Sequence: NM_004613.4). Transfection was completed following manufacturer’s instructions. Briefly, 30 nM of scrambled siRNA-negative control and TGM-2 siRNA (NM_004613) were diluted in basal DMEM and incubated with HiPerfect transfecting reagent (Qiagen) at room temperature for 10 min before being added dropwise to cells. Cells were then placed back into the incubator and monitored for a further 48 h. Knockdown efficiency was determined using q-PCR to confirm that the gene of interest had achieved a knockdown efficiency of at least 60 percent.

### 2.12. Transfection

Gene silencing was used to confirm the interaction between TGM2, mAChR2, Col5a2, Col1a1, and Co1a2. Prior to transfection, 3.5 × 10^4^ MSF cells from homozygous TGM-2 KO mice and control wild-type mice were plated in 12 well plates. All siRNA used were purchased from Qiagen. Transfection was completed following manufacturer’s instructions. Briefly, 30 nM of various siRNA; MAPK1 siRNA (positive control) (Genbank accession number NM_011949), scrambled siRNA-negative control, M2 siRNA (NM_203491), and Col5a2 (NM_007737) were diluted in basal DMEM incubated with HiPerfect transfecting reagent (Qiagen, Hilden, Germany) at room temperature for 10 min before being added dropwise to cells. It was then monitored for the next 48 h before typsinising the cells for q-PCR to confirm that the gene of interest had achieved a knockdown efficiency of 60 percent.

### 2.13. In Vivo Drug Treatment

To determine the effect of muscarinic antagonists on TGM-2 in the development of myopia, mice were treated with the following drugs two weeks after wearing minus lenses. Mice were divided into five groups: the first group received one daily drop of 0.1% atropine sulfate (Sigma-Aldrich, St. Louis, MO, USA), the second group received one daily drop of 0.1% himbacine (Sigma-Aldrich, St. Louis, MO, USA), the third group received one daily drop of 0.1% AFDX-116 (Sigma-Aldrich, St. Louis, MO, USA), the fourth group received one daily drop of 0.1% darifenacin (Sigma-Aldrich, St. Louis, MO, USA), and the fifth group received 1 daily drop of 0.1% carbachol (*n* = 30 mice in each group; (Sigma-Aldrich, St. Louis, MO, USA). Topical applications were administered to each of the two eyes at the same time each day (approximately 9:00 AM) commencing on postnatal day 24 (after 2 weeks of spectacle lens treatment). Eyes were examined daily, and no infections were found. This treatment schedule continued for four weeks, starting on postnatal day 24 and continuing until postnatal day 52. All measurements were taken at postnatal day 52, the equivalent of 6 weeks of spectacle lens wear. The cornea, retina, and scleral tissues were collected for qRT-PCR gene expression analysis.

### 2.14. Quantitative Real-Time Comparative Polymerase Chain Reaction (qRT-PCR)

Total RNA was isolated from cultured SFs and scleral tissue from human and mice (TGM-2 KO, WT mice and human SFs, *n* = 3 sets; 1 × 10^6^ cells/set; *n* = 6 sclera from each strain and *n* = 3 human sclera) using TRIzol reagent (Invitrogen Life Technologies, Carlsbad, CA, USA) in accordance with the manufacturer’s instructions. Genomic DNA was removed by digestion with DNase I (Amp Grade; Invitrogen-Gibco, Waltham, MA, USA) for 15 min at RT. One microgram of total RNA was reverse-transcribed with random hexamers by using a first-strand cDNA synthesis kit (Invitrogen-Gibco). qPCR was performed in a 384-well plate format using a Roche 480 LightCycler Detection System (Roche Applied Science, Mannheim, Germany) with efficiency-corrected software 4.0. PCR was performed using 50 ng of cDNA of each sample. The pre-validated hydrolysis probes for TGM-2, mAChR1-5, Col1a1, Col1a2, and Col 5a2 were from human and mouse universal probe library (Roche), and the primers for human and mouse were the same as those mentioned previously [18,19]. Glyceraldehyde 3-phosphate dehydrogenase (GAPDH) Internal Standard (Roche) was used as an endogenous control. To standardize and evaluate scleral gene expression, aliquots of the same cDNA (50 ng) preparation were used as templates in all PCR reactions. The data were analyzed by using the comparative CT (ΔΔCT) method, and a 1.5-fold change was considered significant.

### 2.15. Data Analysis

Differences in refractive power and axial length between eyes at each timepoint were calculated. Independent *t*-tests were used to compare experimental and control eyes in WT and TGM-2-deleted mice. Statistical comparisons between experimental groups were conducted using Student’s *t*-test or one-way ANOVA (SPSS, Chicago, IL, USA). The Mann–Whitney U-test was used to determine differences between groups. A significance level of *p* < 0.05 was used. Data are presented as means ± standard deviation.

## 3. Results

### 3.1. Up-Regulation of TGM-2 in WT Experimental Myopic Sclera

A previous study showed that TGM activity was mainly localized in the episcleral vessel walls within the sclera [14]. Myopia is a refractive condition in humans that involves the abnormal remodeling of the sclera due to improper visual stimuli [19,20]. In a mouse model of myopia induced by the wearing of a negative lens on one eye, we observed high levels of TGM-2 protein staining in the sclera (Figure 1A) and elevated TGM-2 transcription levels (Figure 1B) and protein levels (Figure 1C) in myopic sclera compared to the control sclera (normalized to the GAPDH). In comparison to the control eye, the spectacle lens treatment applied to wild-type mice significantly reduced scleral thickness (Figure 1A). Our results evidence the change in the regulation of TGM-2, an extracellular matrix-associated cross-linking enzyme, in experimental myopia in the mouse sclera. Thus, we hypothesize that the regulation of TGM-2 is necessary for the growth of the sclera in experimental myopia (Figure 1A–C).

### 3.2. TGM-2 Gene Protected against Development of Experimental Myopia

Our findings demonstrate that homozygous TGM-2 mutant mice remained hyperopic at week 6 following negative lens-induced myopia induction, in contrast to wild-type (WT) mice (Figure 2A). Notably, plano-powered lenses did not induce myopia in WT mice. While axial length elongation was significantly higher in WT mice at week 6, negative lens treatment did not result in significant axial length elongation in TGM-2 mutant mice compared to the uncovered eyes of the same animals (Figure 2B). No significant differences were observed in anterior chamber depth between minus-lens treated eyes and contralateral eyes in either strain (Figure 2C). Varying vitreous chamber depths were observed in the minus-lens-treated eyes of WT mice at different induction durations (*p* < 0.06 at 2 weeks induction; *p* < 0.05 at 4 weeks induction; and *p* < 0.01 at 6 weeks induction, *n* = 60), whereas no significant differences were observed in myopia-induced TGM-2 KO mice when compared to contralateral eyes (*p* = 0.15, *n* = 60) (Figure 2D). Similar results were obtained when diffuser lenses were used instead of negative lenses. Overall, our data suggest that TGM-2 plays an indispensable role in the development of myopia in this animal model by regulating axial elongation and scleral growth. Specifically, the homozygous silencing of the TGM-2 gene protected against myopia development by slowing axial elongation and scleral growth. Since muscarinic receptor antagonists reduced axial length elongation in most animal models of myopia and in human childhood myopia, we were interested in determining if the action of these drugs in myopia involved TGM-2. Through GTPase, both muscarinic receptors and TGM-2 signals can increase calcium, with muscarinic receptors acting on ion channels and TGM-2 acting via phospholipase C. In the sclera-derived fibroblasts from wild-type mice, treatment with the pan-muscarinic antagonist atropine induced a reduction in TGM-2 transcription levels (Figure 2E), protein levels (Figure 2F, top), and cross-linking activity (Figure 2F bottom), whereas the pan-muscarinic agonist carbachol had the opposite effects (Figure 2E,F).

### 3.3. Myopia Inhibition via Muscarinic Receptor

Atropine sulfate is a competitive antagonist of the muscarinic acetylcholine receptors (M1–M5) that is found in various tissues of the body. It blocks the binding of acetylcholine to these receptors, thereby inhibiting the parasympathetic (cholinergic) nerve impulses. It produces a broad range of pharmacological effects, including reducing salivary and respiratory secretions, dilating pupils, and relaxing smooth muscles.

Himbacine is a muscarinic antagonist that selectively targets the M2 and M3 subtypes of muscarinic receptors. By blocking these receptors, himbacine inhibits the effects of acetylcholine, which is the primary neurotransmitter of the parasympathetic nervous system. This blockade results in the relaxation of smooth muscles, reductions in glandular secretions, and other pharmacological effects like those of other muscarinic antagonists.

AFDX-116 is a selective antagonist of the M2 and M3 subtypes of muscarinic receptors. It blocks the binding of acetylcholine to these receptors, preventing the activation of downstream signaling pathways. This blockade leads to a reduction in parasympathetic activity, resulting in the relaxation of smooth muscles, inhibition of glandular secretions, and other effects associated with muscarinic antagonism.

Darifenacin is a selective antagonist of the M3 subtype of muscarinic receptors, which are primarily found in the bladder smooth muscle. By blocking these receptors, darifenacin reduces the bladder’s contractions, thereby increasing its capacity and reducing the frequency of urination. It is primarily used to treat overactive bladder symptoms.

Carbachol is a direct-acting cholinergic agonist that mimics the effects of acetylcholine. Although it is not a muscarinic antagonist, it is mentioned here for reference and was used as a positive control. Carbachol stimulates muscarinic receptors, leading to the activation of parasympathetic responses. It is used in ophthalmology to constrict the pupils during certain procedures and in the treatment of glaucoma by reducing intraocular pressure.

The fact that muscarinic receptor antagonists have been effective in reducing axial length elongation in animal models of myopia and in childhood myopia in humans led us to investigate whether their action involved TGM-2. Our previous research [17] showed that treatment with the pan-muscarinic antagonist atropine reduced TGM-2 transcription levels, protein levels, and cross-linking activity in sclera-derived WT fibroblasts, while the pan-muscarinic agonist carbachol had the opposite effect. In this study, we treated WT myopic murine eyes with different muscarinic antagonists for four weeks, following the same protocol outlined in previous studies. Our results indicate that atropine and 4-DAMP treatment significantly reduced TGM-2 transcription and proteins in sclera tissue compared to the untreated eyes (*n* = 36 eyes in each drug, *p* = 0.004 and *p* = 0.0006, respectively; Figure 3A), while himbacine and darifenacin had the most significant effect (*n* = 36 eyes in each drug, *p* = 0.00003 and 0.0002, respectively, Figure 3B). These findings suggest that TGM-2 mediated the effects of the mAChR2/3 antagonist in this myopia model to reduce myopia [18,19,20].

### 3.4. Effect of Muscarinic Agents or TGM-2 Inhibitors on Cultured Scleral Fibroblast

#### 3.4.1. SF Cell Growth

In this experiment, cultured SF cells at passage 4 were treated with different muscarinic agents (0.1% atropine sulfate; 0.1% Himbacine; 0.1% AFDX-116; 0.1% Darifenacin; 0.1% carbachol) at varying concentrations for 5 days. The SF cell growth was monitored by Xcelligence cell impedance, and TGM-2 transamidase activity was measured using a TGM-2 calorimetric assay. Our results showed that specific muscarinic receptor antagonists reduced the transamidase activity of endogenous cellular TGM-2 in a concentration-dependent manner in both mouse (Figure 4A) and human SFs (Figure 4B). Atropine and himbacine were found to be the most effective inhibitors of TGM-2 activity in both cells (*n* = 3, *p* < 0.0001).

#### 3.4.2. Cell Proliferation: Doubling Time of WT Sclera Fibroblast versus TGM-2 KO Fibroblast

The doubling time between TGM2 KO and WT SF cells were compared (Figure 5). Between 2 and 24 h, it was observed that TGM-2 KO SF cells had a higher doubling time of 24 h as opposed to 16 h in WT cells (Figure 5A), indicating that TGM-2 KO cells had a slower growth rate (Figure 5B).

SF cells derived from TGM-2 mutant mice showed a significant reduction in cell growth after 7 h, and this reduction became more significant over time compared to WT SF cell growth (Figure 5C). Between 24 and 48 h, little difference between the doubling time for TGM-2 KO SF and WT cells was noted. Lastly, our doubling time comparison for intervals between 96 and 120 h (approximately between day 4–5), showed the greatest difference in doubling time between the two cell types (82 h for TGM-2 KO and 48 h for the WT SF). Our doubling time comparison between the two cell types showed this at most time interval comparisons.

Human scleral fibroblasts were silenced with TGM-2 SiRNA to confirm the regulation of TGM-2 in scleral growth. The TGM-2 SiRNA-treated SF had a greater doubling time than the untreated group and exhibited a slower rate of growth (Figure 5D).

Our recorded knockdown efficiency values confirmed that TGM-2 achieved a knockdown efficiency of 60 percent after knockdown with TGM-2 siRNA in human SF cell (Figure 5E). The results confirmed that TGM-2 is important for scleral fibroblast growth, and this was significantly reduced with TGM-2 SiRNA treatment compared to the untreated group.

#### 3.4.3. TGM-2 Down-Regulation Profile in SFs

Wild-type mouse SFs treated with the various transglutaminase inhibitors down-regulated TGM-2 expression to various extents by the end of day 3 (Figure 6).

##### TGM-2 Down-Regulation in MDC-Treated Cells

The effect of the four TGase inhibitors regarding down-regulation was least obvious in the cells that were treated with various concentrations of MDC. It was shown that a concentration of 10 μM had the greatest effect on the down-regulation of TGM-2 (−1.9-fold), followed by cells that were treated with 25 μM and 5 μM, respectively. In the first 2–24 h, MDC inhibitor treatment was observed to have little effect on the WT cells. No significant difference between the MDC-treated and WT cells was observed. The doubling time was 14, 15, 16 h for 5 μM, 10 μM, and 25 μM of MDC, respectively. A significant difference in doubling time was observed at 24 and 48 h. The cells treated with MDC experienced an increase in doubling time at 26.5, 28, and 27 h for the concentrations of 5 μM, 10 μM, and 25 μM, respectively. In comparison, the rate of growth slowed in the time intervals between 48 and 72 h. The cells treated with 5 μM and 10 μM MDC experienced a slight increase in doubling time compared to the WT control in a dose-dependent manner.

However, when the doubling time was measured between 96 and 120 h, the doubling time of the MDC-treated cells was lower than that of the WT cells. The doubling time for the 25 μM MDC-treated MSF was 46 h, which was lower than the WT control at 50 h. This decrease in doubling time co-relates with the dose response curve at 120 h, whereby the cell index increases in response to MDC rather than decreases (Figure 6A).

##### TGM-2 Down-Regulation in Z006-Treated Cells

On the other hand, the Z0006-treated cells were treated with higher concentrations compared to MDC. This decision was based on the information presented by Siegel et al. [21], which essentially indicated that a concentration of 40 μM was necessary to stimulate and inhibit intracellular TGM-2 and that an excess of 80 μM would generate a cytotoxic effect in human cells. Hence, concentrations of 40 μM, 55 μM, and 70 μM were used. Based on our results, 70 μM resulted in the greatest inhibition of TGM-2 gene expression, with a −13.7-fold reduction. Unfortunately, cells treated with 55 μM concentration resulted in an up-regulation of TGM-2 expression.

The doubling time measured between 2 and 24 h for the cells treated with Z006 ranged from 13 to 17 h. In comparison, the doubling time of the Z006-treated cells was lower than the WT vehicle-treated sample at 16 h. On most occasions, between 24 and 72 h, the doubling time of the Z006-treated cells was higher than that of TGM-2 KO. A drastic increase in doubling time was noted between 24 and 48 h. The doubling time was greater at 30–34 h for the Z006-treated cells compared to the WT cells, which had a doubling time of 24 h. The same was observed between 48 and 72 h, whereby the doubling time ranged from between 31 and 38 h. These values demonstrate a significant increase compared to the WT cells’ doubling time of 26 h.

However, when the doubling time was measured between 96 and 120 h, it was observed that only the doubling time of Z006 at a concentration of 70 μM was greater than that of the WT control at 72 h. This value was the highest, although this value was slightly lower than that of the TGM-2 KO cells.

At 70 μM, Z006 appeared to be the most effective in increasing the doubling time without leaving any physical signs of cell toxicity via changes to cell morphology (Figure 7B).

##### TGM-2 Down-Regulation in D003-Treated Cells

Treatment with D003 at a concentration of only 10 μM was sufficient enough to generate an 8.1-fold reduction in TG2 expression. However, treatment at a concentration of 25 μM resulted in a lower reduction (−4.7-fold). The down-regulation of TG2 by D003 at 10 μM was relatively significant (−8.1-fold) compared to the cells that were treated with Z006 at 70 μM, though a much higher concentration was required to illicit a comparable effect. On the other hand, a concentration of 1µM resulted in a 1.6-fold down-regulation of TG2.

In general, a steady increase in doubling time was recorded with increased incubation time following treatment with D003. The cells treated with D003 for all three concentrations in the first 2–24 h had a doubling time of 12–13 h, while for WT vehicle-treated cells, this figure was 16 h. Between 24 and 48 h, only the cells treated with 25 μM D003 had a doubling time of 28 h, which was greater than the WT control (doubling time of 24 h). The doubling times for cells treated with D003 at a concentration of 1 μM and 10 μM were 27 and 23 h, respectively. A similar pattern was also observed when the doubling time was measured between 48 and 72 h. At a concentration of 1 μM, the doubling time was 28 h, but it increased to 32 h when 10 μM was added. However, the doubling time of the cells treated with 25 μM D003 was 26.5 h, while for WT, this figure was 26 h.

Lastly, when the doubling time was measured between 96 and 120 h, apart from at a concentration of 1 μM, there was a general increase in doubling time (64.5 h) compared to the WT group (50 h). Lower concentrations of 10 and 25 μm of D003 were enough to increase the doubling time to 60 h (Figure 6C).

##### TG2 Down-Regulation in KCC009-Treated Cells

Lastly, the cells treated with KCC009 had the second-least amount of influence on TG2 down-regulation. At 10 μM, a down-regulation of 3.4-fold was recorded. At the same concentration, that of D003 was −8.1-fold, and for MDC, this figure was −1.9-fold. The cells treated with KCC009 at a concentration of 25 µM had a lower reduction (−2.3-fold), and at a concentration of 50 µM, a 1.7-fold boost in up-regulation was observed (Figure 6D).

There was no changes observed in terms of cell morphology. However, the cells treated with higher concentrations of MDC, a TGM competitive inhibitor, caused cell death and morphological changes. IC_50_ values were calculated based on dose response curves from 0 h to 120 h. The IC_50_ value for MDC was 3.0 × 10^−3^; KCC009—2.8 × 10^−5^; Z006—4.9 × 10^−5^, and D003—9.3 × 10^−6^. D003 appears to have the lowest IC_50_ value, suggesting that it might be the most effective inhibitor.

### 3.5. Expression of Muscarinic Receptors in TGM-2 KO Mice Sclera

The levels of TGM-2 mRNA were measured in the sclera of mAChR1-5 knockout mice. M1, M4, and M5 knockout mice had higher levels of TGM-2 compared to M2 and M3 knockout mice, and there was no significant difference in all WT mice, as seen in Figure 7A. The differential gene expression was significant in M2 and M5 knockout mice sclera. This pattern was also observed in the protein levels, as shown in Figure 7B. These findings support the notion that M1, M4, and M5 mutants have a greater axial growth than M2 and M3 mutants [16]. These results demonstrate that TGM-2 expression is up-regulated in spectacle lens-induced myopic sclera, which is associated with axial length elongation and scleral thinning (Figure 1A indicated by the arrow), compared to the contralateral control sclera. These findings suggest that the interaction between TGM-2 and muscarinic receptors may be implicated in scleral remodeling.

#### Transfection of Scleral Fibroblast with Col5a2, M2 in Both WT and TG2 KO Scleral Fibroblasts

In order to study the gene–gene interactions between TGM-2, mAChR2, Col1a1, Col1a2, and Col 5a2, we silenced Col 5a2 and mAChR2 in both TGM-2 KO and WT cells (Figure 8).

### 3.6. Silencing of TGM-2 Leads to a Decreased Expression of mAChR2

When TG2 was silenced, it was observed that M2 is down-regulated by two-fold (Figure 8A). Hence, we suggest that there could be a possible interaction between the two genes. It was also noted that Col 5a2 is reduced slightly (−1.6-fold), while Col 1a1 and Col 1a2 expression is up-regulated slightly (1.3- and 1.5-fold, respectively).

### 3.7. Silencing of mAChR2 Has a Slight Effect on Col 5a2 Expression

On the other hand, to establish a relationship between TGM-2, mAChR2, and the collagen genes, we silenced mAChR2 in WT cells (Figure 8B). Little effect over the expression of Col 5a2 was observed when M2 was silenced. However, it was noted that there was an increase in the expression of Col 1a1 and Col1a2 (+1.4-fold).

Additionally, when the mAChR2 gene was silenced in the TGM2 KO MSF, a slight reduction in Col 5a2 expression was observed (Figure 8C). However, a greater level of up-regulation regarding Col 1a1 and Col 1a2 was observed (1.2- and 2-fold, respectively), as shown in Figure 8D. Hence, we determined that there could be possible interactions between TGM-2 and mAChR2, as the silencing of TGM-2 and mAChR2 led to a greater expression of Col 1 mRNA.

### 3.8. Silencing of TGM-2 and mAChR2 Leads to Col 5a2 Down-Regulation

When TGM-2 and mAChR2 were silenced and compared against a WT MSF sample, a greater down-regulation of Col 5a2 was observed (−1.4-fold) (Figure 8E). However, the increase in Col 1 expression was comparatively low (1.1- and 1.3-fold for Col 1a1 and Col 1a2, respectively). This could be due to the short silencing interval, which was employed to induce a greater expression of Col 1.

### 3.9. Silencing of Col 5a2 Leads to Increased Col 1 mRNA Expression

When Col 5a2 was silenced in the WT SF, it was observed that mAChR2 was down-regulated by −1.4-fold (Figure 8C). On the other hand, Col 1a1 and Col1a2 were up-regulated slightly (+1.3- and +1.5-fold, respectively).

On the other hand, the silencing of Col 5a2 in TGM-2 KO MSF yielded minimal change in the expression level of M2 (i.e., a change of 1.1-fold); however, the expression of Col 1a1 and Col 1a2 was up-regulated to a greater extend (1.2- and 2-fold, as shown in Figure 8F).

Also, when TGM-2 and Col 5a2 were silenced and compared against a WT control (Figure 8G), a greater up-regulation of Col 1 was observed, suggesting that Col 5a2 may interact with TGM-2 to regulate Col 1 expression.

Clearly, it can be concluded that the silencing of Col 5a2 is usually accompanied by the up-regulation of Col 1, with a greater effect being evident when TGM-2 is silenced.

## 4. Discussion

Despite the fact that there is more than one type of human myopia (late-onset and developmental) and animal myopia (lens-induced, form deprivation, etc.), it seems that all myopia is mediated by axial elongation, resulting in a mismatch between the optical refractive properties of the anterior segment relative to the position of the photoreceptors on the retina. Prior studies have reported that axial elongation could be mediated by alterations in the connective tissues of the sclera, a fibrous coat in the eye. Hence, the modulation of connective tissue molecules in the sclera may represent a strategy for arresting all types of myopia development, regardless of the initiating stimulus [22]. This also suggests that applying an animal model of myopia to investigate scleral remodeling is appropriate in order to facilitate the discovery of useful anti-myopiagenic drugs for human use.

When the TGM-2 gene was homozygously silenced, mice were protected against the development of myopia by slowing axial elongation and scleral growth. Additionally, mice with M1, M4, and M5 mutations had higher levels of TGM-2 compared to M2 and M3 mutants, as seen in both the mRNA and protein levels in KO and WT mice. These results suggest that TGM-2 plays a role in myopia development by directly controlling SF cell growth, and blockers of TGM-2 could potentially be used as target drugs to reduce myopia progression. These results correlate with those of published studies [23,24,25,26,27,28,29,30]. Furthermore, TGM-2 was found to interact with mAChRs and other extracellular matrix genes to regulate scleral growth-related changes during myopia development.

### 4.1. Role of TGM-2 in Myopia

Li et al. [14] investigated the involvement of TGM-2 in scleral fibroblast proliferation and migration, which are processes believed to contribute to the axial elongation of the eye. The study found that TGM-2 mediates these processes, suggesting its potential role in the development of myopia. Huang et al. [15] studied the scleral tissues of tree shrews, which were used as an experimental myopia model. They observed the up-regulation of TGM-2 and collagen III in the scleral tissues, indicating their possible involvement in myopia development. Wu et al. [16] utilized a guinea pig model of myopia to investigate the role of TGM-2 in axial elongation. The study demonstrated that TGM-2 is involved in the development of axial elongation, further supporting the idea that TGM-2 plays a role in myopia progression.

The results of the aforementioned studies [14,15,16] and this study’s results collectively suggest that TGM-2 may play a significant role in the development of axial elongation and myopia. TGM-2 appears to be involved in processes such as fibroblast proliferation and migration, as well as extracellular matrix remodeling, which are crucial for the structural changes observed in myopia. The up-regulation of TGM-2 in the scleral tissues of myopic eyes suggests its potential as a therapeutic target for controlling myopia progression. Additionally, considering the results reported in the present study that pertain to the effects of TGM-2 inhibition or modulation in other animal models and subsequent clinical trials could offer a better understanding of its potential as a therapeutic target for myopia management.

### 4.2. Interaction between Transglutaminase and Muscarinic Receptors

Myopia is a refractive condition in humans that is characterized by the abnormal remodeling of the sclera in response to inappropriate visual stimuli. In a mouse model of myopia induced by wearing a negative lens on one eye, we observed that scleral fibroblasts increased staining for TGM2 protein, and TGM2 RNA levels were higher in the myopic eye than in the control eye.

When fibroblasts were cultured from the scleral explants of these mice, treatment with atropine, a pan muscarinic receptor antagonist, led to a decrease in TGM2 protein and RNA levels. Previous research on the muscarinic pathway in neuroblastoma cells has shown that TGM activity increases after agonistic stimulation [31,32,33,34]. These findings suggest that the manipulation of TGM via muscarinic receptors could be a viable method for intervening in scleral remodeling and potentially slowing the progression of myopia.

Our study, on the other hand, offers insight into the effects of TGase inhibitors on intact cells in vitro without cell disruption to stimulate physiological conditions. In this study, we tested the potency and the efficiency of four commercially available TGase inhibitors. We used various methods involving cell impedance technologies and qPCR techniques to identify the most effective TGase inhibitor and to determine the concentration that would reduce the proliferative potential of scleral fibroblasts without causing cellular cytotoxicity.

TGM-2 plays a role in many important biological processes and the pharmacological inhibition of TGM-2 might have undesirable effects. The results from our xCelligence study confirmed that slower cell growth rates were recorded when MSF were treated with TGase inhibitors. Based on our q-PCR results, we noted that 70 μM of Z006 was the most effective concentration with regards to inhibiting TGM-2 expression. This was followed by D003, KCC009, and MDC. IC_50_ values, together with dose response measurements taken at various time points, were used as a measure of compound potency. As demonstrated in our study, D003 had the lowest IC_50_ value (9.3 × 10^−6^). Considering that D003 is a weaker blocker of TGM-2 activity compared to Z006, it was not surprising that treatment with D003 resulted in a lower decrease in the cell index and yielded a lower doubling time compared to Z006. In fact, the presence of a DON (6-diazo-5-oxo-norleucine)-reactive group on Z006 made it a more selective inhibitor; hence, despite having a higher IC_50_ value, Z006 most effectively decreased cell proliferation rates, particularly at 70 μM. In our study, we noted that the IC_50_ values taken at 24, 48, 72, 96, and 120 h for various inhibitors varied drastically. We suggest that, since IC_50_ may be highly time-dependent, if values are recorded while the reaction is not complete, they may differ greatly.

### 4.3. Extracellular Matrix Interactions

Transglutaminase plays a crucial role in regulating the fundamental functions of cell adhesion and spreading. The absence of TGM-2 in primary fibroblasts from knockout mice results in reduced attachment to culture vessels. Cell–matrix interactions are essential for various cellular processes, including spreading, migration, and extracellular matrix organization [35,36].

Transglutaminase is known to interact with several components of epithelial basement membranes and dermo–epidermal complexes, including integrin α6β4 and BP180/collagen XVII, the laminin–nidogen complex, osteonectin, fibronectin, collagen type VII, fibrillin 1, and the microfibril-associated glycoprotein precursor (MAPG1), as previously noted elsewhere [37,38,39].

### 4.4. Regulation of Transcription

Modifying transcription factors within cells can have a significant impact on the expression of related genes [40,41]. Treatment with purified liver transglutaminase (TGM) increased the binding activity of Sp1, a transcription factor rich in glutamine, to its target DNA sequence, as confirmed by gel electrophoretic mobility shift assay [42]. Furthermore, 293T cells over-expressing TGM2 demonstrated an increase in Sp1 protein levels. TGM2’s transamidation activity may also cross-link Sp1 from nuclear extracts in vitro. Additionally, TGM has been associated with the transcription of papillomavirus genes [43,44,45].

### 4.5. G-Protein Signaling

TGM-2, also known as tissue transglutaminase, is expressed widely throughout the body. Its primary function is transamidation, which requires calcium as a cofactor [46]. It is interesting to note that retinoic acid increases its transcription. Although it is believed to have many functions, TGM-2 appears to play a role in wound healing, apoptosis, and the development of the extracellular matrix [47,48,49]. Additionally, in the myopic sclera, TGM-2 exhibits GTPase activity and functions as a G protein in signaling processes when GTP is present. The findings indicate that mutant mice with the M1, M4, and M5 mutations had a higher axial growth than those with the M2 and M3 mutations. These results suggest that the interaction between TGM-2 and muscarinic receptors may be involved in scleral remodeling.

The TGM2 protein is the oldest known bête noir of the TGM family and is unique in its ability to function as a G protein, as recently reviewed. The heterotrimeric G protein subunit is called Gh, which is similar to the well-known Galpha subunit of large G proteins. TGM2, acting as a G protein, is associated with alpha1B-adrenergic receptors and may activate downstream signaling pathways, including phospholipase C activation. This pathway can increase cytosolic calcium levels and activate the transamidation function of TGM2 since phospholipase C can generate inositol triphosphate, a potent stimulator of calcium channels. The Gh signaling of TGM2 is independent of the requirement for cysteine in the active site and the transamidase function [50]. In addition to GTP, other molecules such as intracellular zinc ions and nitric oxide also inhibit the cross-linking activity of TGM2. While the G protein signaling may be the default function of cellular TGM2 in physiological conditions, the transamidase function may be more prominent in pathological and severe conditions because the calcium concentration required for activating transamidation is relatively high compared to normal cytosolic calcium levels [51,52]. It is important to note that GTP does not affect the transamidase function of other TGM, such as TGM3.

In the absence of TGM2, fibroblasts can still attach to the extracellular matrix, but they exhibit deficient spreading and migration [53]. Moreover, they show defects in the turnover of focal adhesions and formation of stress fibres. These motility and adhesion kinase phosphorylation defects are unrelated to the transamidase function but are linked to the G protein function of TGM2 [54,55]. Furthermore, TGM2 is necessary for the activation of MMP2 by membrane type 1-MMP. If cross-linking by TGM2 is disturbed, collagen matrix contraction is reduced, and gelatinase activation is compromised. Therefore, both the G protein and transamidase functions of TGM2 independently contribute to scleral remodeling processes.

### 4.6. Transglutaminase as a Target for Drugs

Ongoing research should focus on developing newer and more effective inhibitors of TGM. Previous inhibitors, such as the cell-permeable dansylcadaverine and 3-halo-4,5-dihydroisoxazoles, have been used. However, more recent inhibitors have been discovered through various means, which not only reduced TGM-2 but also other factors such as MMP9, TIMP1 and 2; TNF-α and β, which increases cancer cells’ susceptibility to chemotherapy and inflammation [56,57,58,59,60,61].

Furthermore, researchers have found that the administration of cystamine to mice leads to a decrease in anti-cardiolipin autoantibodies [62]. This discovery indicates that anti-TGM approaches could be a promising treatment option for systemic lupus erythematosus, a severe systemic autoimmune condition [63]. Moreover, a recent study has demonstrated that an octapeptide possessing anti-TGM properties had a beneficial impact on guinea pigs by mitigating lung inflammation. Additionally, a recent study reported that a TGM-2 inhibitor protected kidney cells from the accumulation of ECM molecules, apoptosis, and inflammatory responses [64]. Anti-muscarinic drugs were able to change collagen and other structural molecules in animal models of myopia, establishing a link to scleral connective tissue in these pathways. However, even though muscarinic receptor subtypes may be involved in scleral remodeling, we do not know of their exact downstream molecules. Both TGM-2 and muscarinic receptors have profound effects on the structural molecules of the ECM. Overall, these findings suggest that TGM-2 may be a mediator of muscarinic receptor signaling (Figure 9 and Figure 10) and a more direct modulator of connective tissue.

Further research is needed to gain a deeper understanding of the molecular mechanisms underlying TGM-2’s involvement in myopia development. Investigating the signaling pathways and interactions that regulate TGM-2 activity in the context of myopia could provide valuable insights into potential therapeutic interventions. Additionally, exploring the effects of TGM-2 inhibition or modulation in animal models and clinical trials could offer a better understanding of its potential as a therapeutic target for myopia management.

## 5. Conclusions/Significance

Several recent studies have explored the effectiveness of atropine in managing myopia progression. Although the involvement of muscarinic receptor subtypes in scleral remodeling is recognized, the precise downstream molecules remain unknown. Our research has revealed that transglutaminase (TGM)-2, a molecule previously associated with wound healing and cell migration, is up-regulated in the myopic murine sclera compared to the control. Additionally, we found that the gene expression of TGM-2 is down-regulated by atropine, an anti-muscarinic drug. TGM-2 has also been implicated in regulating extracellular matrix molecules such as fibronectin and collagen. Thus, we recommend investigating the central role of TGM-2 in the development of experimental myopia in mice by inducing myopia in TGM-2-deleted mice and mice treated with TGM inhibitors. We obtained supporting data in the form of TGM-2 associated transcripts and proteins in the sclera, as well as the growth-related changes regarding these molecules in TGM-2-deleted mice. Furthermore, we examined the role of TGM-2 in muscarinic receptor signaling by using TGM inhibitors in mice with deleted muscarinic receptor subtypes prior to inducing myopia. If TGM-2 or its associated molecules are found to contribute to scleral elongation in experimental myopia, they could serve as the basis for novel pharmacological interventions for human myopia.

These findings provide confirmation that TGM-2 plays a critical role in the development of myopia in mice by regulating the growth of SF cells, thereby making it a potential target for drugs aimed at reducing myopia progression. Furthermore, the present study demonstrated that TGM-2 interacts directly with muscarinic receptors and other extracellular matrix genes to control changes in scleral growth that occur during myopia development. These results suggest that atropine, which can be directly or indirectly controlled by TGM-2, is the most promising drug for myopia in children. Based on these findings, we hypothesize that drugs targeting TGM-2 and/or mAChR2/3 could be beneficial in reducing myopia progression while overcoming the side effects of atropine. The significance of these results is that the targeted intervention of transglutaminase may be sufficient to alter the course of myopia. Such specific target drugs could directly modulate scleral remodeling, which occurs during myopia development or progression.

Our goal is to reduce the side effects associated with atropine (the current standard of care for school myopes) by elucidating the molecular mechanism of transglutaminase-2 and muscarinic cholinergic receptors in experimental myopia. This understanding will enable us to develop new drugs—either as monotherapies involving low doses of TGM-2 inhibitors or as combination therapies comprising low doses of TGM-2 inhibitors and atropine—to minimize potential side effects. Prior to translating these interventions to human trials in the future, we will conduct short- and long-term studies in preclinical models to assess safety and efficacy.

## Figures and Tables

**Figure 1 biomolecules-13-01045-f001:**
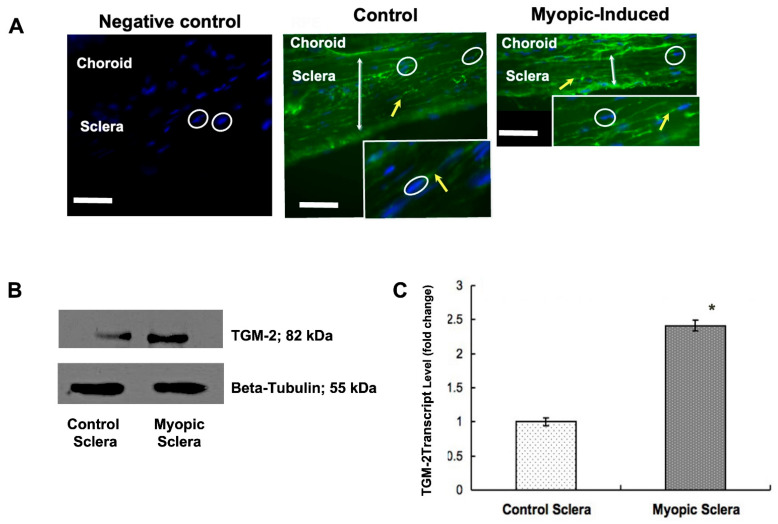
Expression of TGM-2 in eight-week-old murine tissue and cells. (**A**) Immunofluorescent staining images using primary antibody against TGM-2 in wild-type myopic and control murine sclera. Arrow indicates scleral fibroblasts. Scale bar = 50 μm. Inset: enlarged image of fibroblast. (**B**) Results of real-time polymerase chain reaction (PCR) showing TGM-2 transcript levels in wild-type myopic and control murine sclera. Error bars = SD. Student’s *t*-test for independent samples *p* < 0.05. (**C**) Western blot image showing relative protein levels of TGM-2 (top) and tubulin loading control (bottom) in primary fibroblasts cultured from myopic and control murine sclera. GAPDH was used as an internal control, and the message level was normalized with GAPDH housekeeping gene. Error bars = SD. ANOVA *t* test, * *p* < 0.05.

**Figure 2 biomolecules-13-01045-f002:**
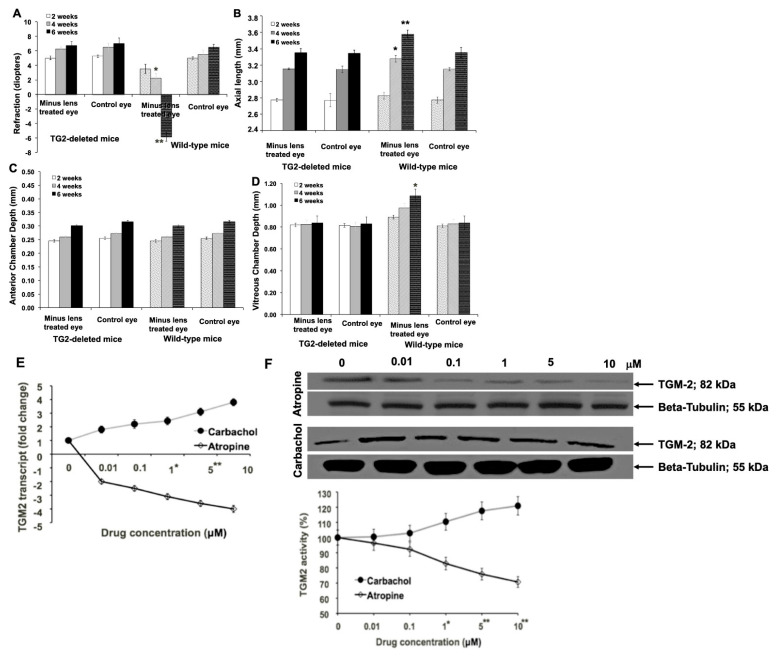
Myopia induction was performed using uniocular -15 diopter negative lenses in wild-type and homozygous TGM2-deleted mice, and results at 2, 4, and 6 weeks are shown. (**A**) Refractive errors were determined via infrared photorefraction. Positive and negative spherical equivalents represent hyperopic and myopic refractive errors, respectively. (**B**) Axial length, (**C**) ACD, (**D**) VCD measurements were determined via optical coherence interferometry. (**E**) Murine scleral-derived fibroblasts were cultured to passage 2 and treated with exogenous atropine or carbachol at different concentrations for 5 days. Following 5 days of treatment, the total RNA was extracted from these cells, and RT-qPCR was performed to determine transcript levels of TGM2. The levels were normalized to GAPDH internal control. (**F**) Total protein was extracted from cells cultured, as described in C, and proteins were detected (**top**) using the primary antibody (Ab421) specific for TGM-2. ß-tubulin was used as a loading control for protein. (**bottom**) Transamidase activity was determined. In all charts, the height of bars or symbols represent mean and error bars SD. *: *p* < 0.05 and **: *p* < 0.01.

**Figure 3 biomolecules-13-01045-f003:**
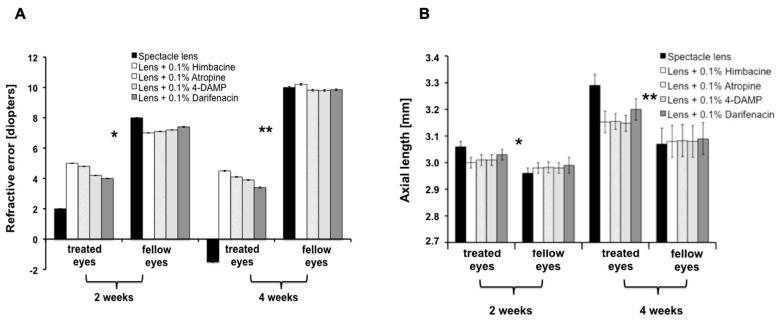
Mice eyes were treated with different muscarinic antagonists (0.1% atropine sulfate; 0.1% Himbacine; 0.1% Darifenacin; 4-DAMP) to determine the drug that is most effective in terms of reducing myopia progression. (**A**) Automated infra-red photorefractor was used to measure the refractive error measurements. The refractive error was shifted from myopic to hyperoic after receiving the drugs. (**B**) Axial length was measured at 2 weeks and 4 weeks after treatment. The axial length was significantly reduced in the drug-treated eyes compared to control and lens-treated eyes. *: *p* < 0.05 and **: *p* < 0.01.

**Figure 4 biomolecules-13-01045-f004:**
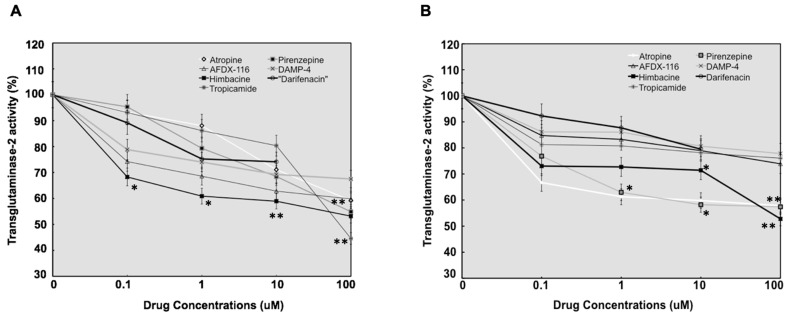
Line graph showing the results of TGM-2 transamidase activity in 100 µg of protein lysate from mouse- (**A**) and human (**B**)-cultured SFs. Values were normalized against control values. The mouse and human SF treated with atropine, pirenzepine, AFDX-116, himbacine, 4-DAMP, darifenacin, and tropicamide at baseline, 0.1, 1, 10, and 100 µM for 5 days. The transamidase activity of endogenous cellular TGM-2 activity was reduced by antagonists’ treatment in a concentration-dependent manner in human and mouse SFs. Moreover, TGM-2 activity was most significantly reduced with himbacine treatment in both cells and also with darifenacin at low concentrations. The values represent the means of three independent samples, and error bars represent standard deviation, *n* = 3. *: *p* < 0.05 and **: *p* < 0.01.

**Figure 5 biomolecules-13-01045-f005:**
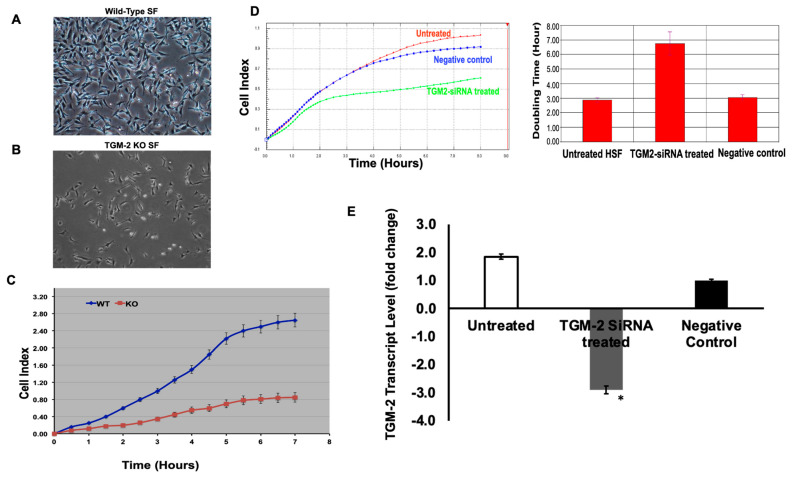
Comparison between WT and TGM-2 KO mice in terms of scleral fibroblast cell growth and proliferation. SF cell growth was monitored via Xcelligence cell impedance. Cell morphology of wild-type mouse (**A**) and TGM-2 KO mouse (**B**) scleral fibroblasts in culture. (**C**) The interaction of cells with the electronic biosensors generates a cell–electrode impedance response that is expressed as cell index, allowing cell numbers to be detected. SF derived from TGM-2 mutant mice cell growth was significantly reduced after 7 h, and this was further reduced with time compared to WT SF cell growth. (**D**) Cultured human SF cells at passage 4 were treated with TGM2-SiRNA, and SF cell growth was monitored via Xcelligence cell impedance. Also, 2000 HSF cells transfected with TGM-2 siRNAs and an untreated control were seeded in 100 μL of media in duplicates in a 96× E-Plate. Cell-sensor impedance was measured every 5 min for the first 2 h, every 15 min for the next 7 h, and every 1 h for the rest of the experiment. An increase in cell index correlates with an increase in cell number. (**E**): Knockdown efficiency of TGM-2 transcription level was quantified by real-time qPCR in TGM-2 siRNA-treated and untreated HSF. * *p* < 0.05.

**Figure 6 biomolecules-13-01045-f006:**
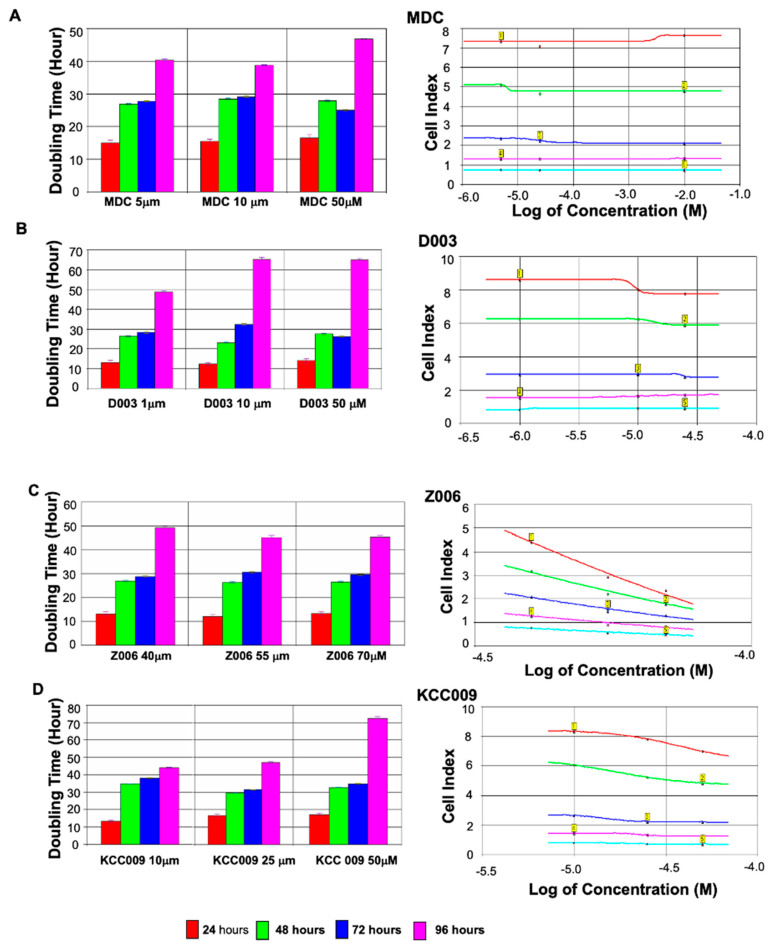
Effect of TGase inhibitors on cultured scleral fibroblast growth. Doubling times of SFs (cell proliferation) that were treated with various concentration of TGase inhibitors. In co-relation with the cells that have been treated with TGase inhibitors, xCelligence was used to determine the growth rates of the cell. Scleral fibroblasts treated with the various TGase inhibitors down-regulated TGM-2 expression to various extents by the end of day 3. Measurement of doubling time at various time intervals (doubling time measurement taken at the respective time intervals). There was a slight increase in doubling time in the MDC-treated SF (**A**). A greater increase in doubling time was observed for the KCC009 (**B**)-treated cells followed by the D003 (**C**)-treated cells. It was shown that R002 (**D**) at 50 μM results in the greatest increase in doubling time. Graphical Representation of dose response curve (cell Index at time point vs. TGase Inhibitors, MDC, Z006, D003, and KCC009). Graph illustrated above indicates the effect of drugs on the cell index, which is a measure of the drugs’ potency (as measured by log of concentration). The cell index was plotted at time intervals of 24 h, 48 h, 72 h, 96 h, and 120 h and the inhibitors’ ability to generate a response is shown. It was seen that at 24 h and 48 h, the TGase inhibitor only has a slight impact on the cell index, apart from for Z006. At 72 h, the effect of the TGase inhibitor on the cell index became more noticeable. A slight slope was observed for the following inhibitors: MDC (**A**), D003 (**B**), and KCC009 (**C**). Z006 (**D**) produced the steepest slope, suggesting that the concentration used were the most effective in reducing cell proliferation ability. At 96 h and 120 h, a greater decrease in cell index was observed for cells treated with KCC009, whereas a slight decrease in the cell index was noted for MDC and D003. Z006 continued to result in the greatest decrease in cell index.

**Figure 7 biomolecules-13-01045-f007:**
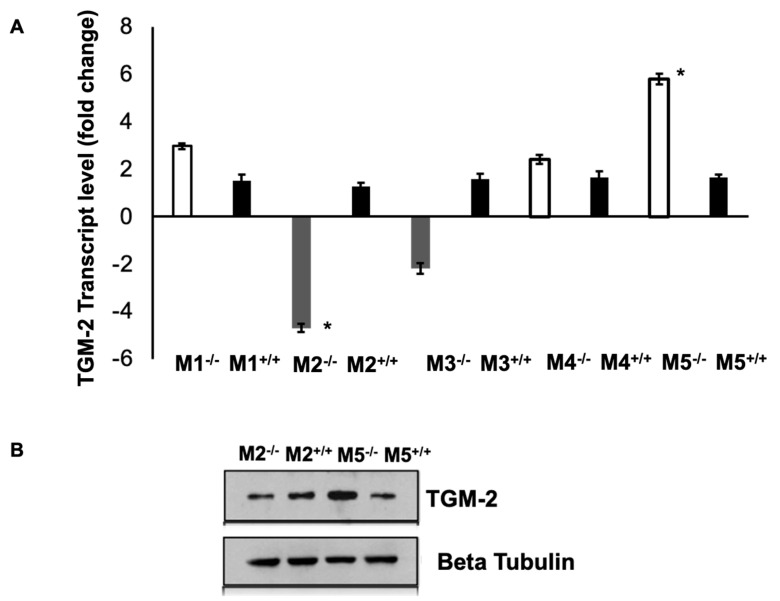
Expression of TGM-2 levels in muscarinic mutant mice sclera. (**A**) Real-time PCR showing relative TGM-2 transcription levels in M1, M2, M3, and M5 knockout and wild-type mice sclera. GAPDH was used as an internal control, and the message level was normalized with GAPDH housekeeping gene. Error bars = SD. ANOVA *t* test *p* < 0.05. (**B**) Western blot image showing relative protein levels of TGM-2 (**top**) and tubulin loading control (**bottom**) in M2 and M5 knockout and wild-type mice sclera. Significance level at * *p* < 0.05.

**Figure 8 biomolecules-13-01045-f008:**
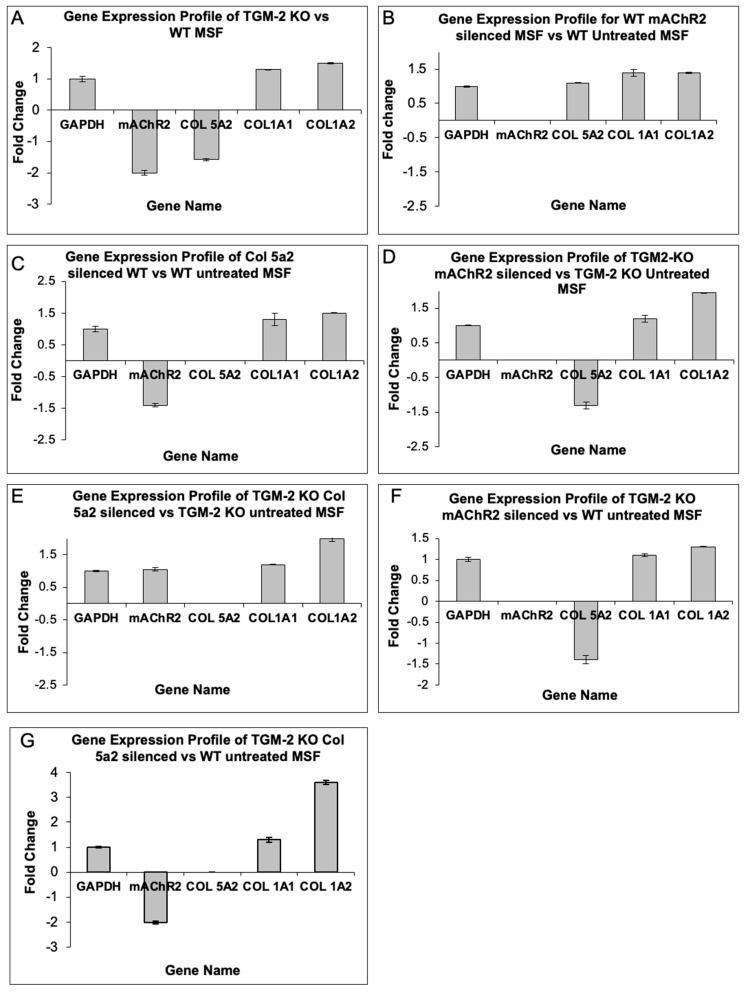
Transfection of TGM-2 KO MSF and WT SF indicated changes to Col 1a1 and Col 1a2 mRNA expression. Silencing of TGM-2 and Col 5a2 led to the greatest increase in Col1 mRNA expression.

**Figure 9 biomolecules-13-01045-f009:**
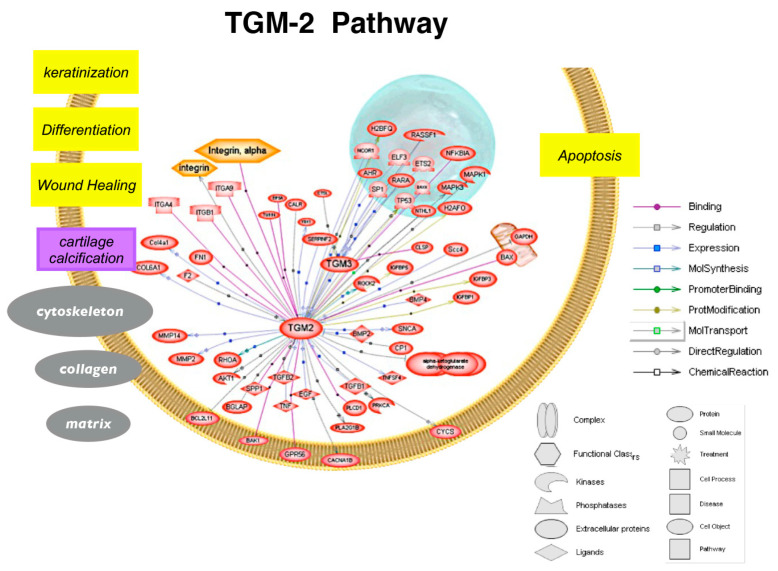
Schematic showing biological processes regulated by TGM-2 pathway (created and analyzed using Pathway Studio 5.0). TGM-2 plays a central role in wound healing, matrix remodeling, and apoptosis.

**Figure 10 biomolecules-13-01045-f010:**
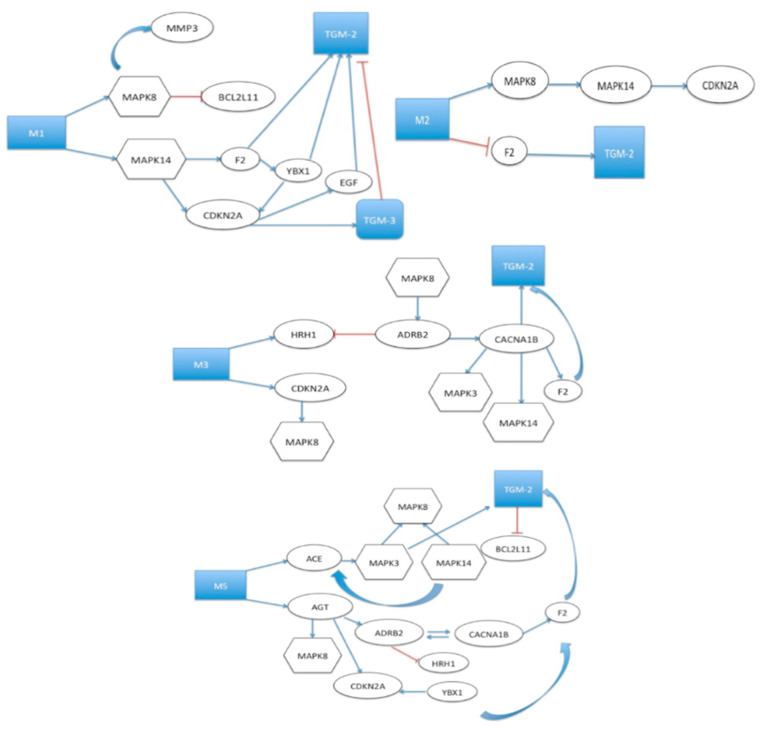
Schematic showing biological processes regulated by TGM-2 as well as interactions between TGM-2 and muscarinic receptor pathway (analyzed using Pathway Studio 6.0). The interaction between TGM-2 and muscarinic receptors becomes involved through the Mitogen-Activated Protein Kinase (MAPK) pathway. These findings suggest that TGM-2 may be a mediator of muscarinic receptor signaling, potentially mediating some of the muscarinic receptor effects on wound healing or scleral remodeling.

## Data Availability

All data generated or analyzed during this study are included in this published article. The datasets used and/or analyzed during the current study are also available from the corresponding author upon reasonable request.

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
