# Peer review of "Molecular Basis of Transglutaminase-2 and Muscarinic Cholinergic Receptors in Experimental Myopia: A Target for Myopia Treatment"

_biomolecules, 2023, doi:10.3390/biom13071045_

Round 1
Reviewer 1 Report
This research manuscript sheds light on potential therapeutic targets for myopia, a common and often progressive eye condition. Targeting TGM-2 and muscarinic receptors may provide new avenues for developing effective interventions to slow or stop the progression of myopia.
The manuscript is written very effectively and thoroughly, and all the necessary information is included. Overall, the manuscript is interesting and has the potential to be accepted. However, the authors must add some important points, which are as follows:
1- One aspect that could be added to the introduction is the current understanding of the underlying mechanisms of myopia, including the role of connective tissue in the sclera and axial elongation. The mentioned study reports evidence that myopia is mediated by axial elongation, resulting from a mismatch between the optical refractive properties of the anterior segment and the position of the photoreceptors on the retina. It is caused by changes in the connective tissues of the sclera.
Therefore, understanding these underlying mechanisms may have important implications for developing effective therapeutic interventions for myopia. Additionally, the introduction may briefly mention the methodological approaches used in the study, including the use of animal models and gene-silencing techniques.
2- From the context provided, it appears that the methods section contains all the necessary information to cover the study. However, some additional details on the experimental setup and animal model could have been included to provide a better understanding of the study. For example, details about the characteristics of TGM-2 mutant mice, the age and sex distribution of the animals used, and the procedure for inducing negative lens-induced myopia in mice could have been provided.
3- Furthermore, more information about the muscarinic antagonists used in the in-vivo drug treatment and their mechanisms of action can be added to provide more contexts to the results obtained.
The Results and Statistics sections are very well presented after the Methods section. However, in terms of the order within the Results section, it appears that findings relating to the TGM-2 gene and the development of experimental myopia are presented before subsequent experiments on gene expression, proteomics and drug treatment. This is a reasonable approach as it follows a logical flow of presenting results based on the research question and the hypothesis being tested. In the statistics section, it appears that appropriate tests were used, including Student's t-test, Mann–Whitney U test, and ANOVA, and these are generally ordered correctly based on the context provided.
Author Response
Thank you for taking the time to review our research manuscript titled "Therapeutic Targets for Myopia: Targeting TGM-2 and Muscarinic Receptors." We sincerely appreciate your thoughtful comments and constructive feedback. We have carefully considered your suggestions and have made the necessary revisions to improve the clarity and comprehensiveness of our manuscript. Below, we address each of your points in detail.
- We agree with your suggestion to include in the Introduction section on the current understanding of the underlying mechanisms of myopia, particularly focusing on the role of connective tissue in the sclera and axial elongation. We have now expanded the introduction to provide an overview of these mechanisms, emphasizing the connection between axial elongation and changes in the connective tissues of the sclera. Additionally, we have briefly mentioned the methodological approaches employed in our study, including the use of animal models and gene-silencing techniques.
Introduction Section Pg 52-80: “Myopia is a condition where the normal process of emmetropization, which occurs within the eye, fails to function properly. Emmetropization involves the detection of defocus by certain cells in the outer retina, such as the amacrine and bipolar cells. This information is then transmitted across the retinal pigment epithelium and choroid through a signal or signals. The scleral matrix, which is the structure surrounding the eye, is also modified, likely through the regulation of proteoglycan synthesis (4). Our understanding of myopia has been improved through clinical experience, longitudinal studies, epidemiological data, and various animal experiments. However, distinguishing between environmental and genetic influences, especially those related to gradual developmental changes, can complicate the matter.
To study emmetropization and the development of myopia, researchers have utilized different animal models, including chicks, rabbits, tree shrews, guinea-pig, mice and macaques (5). These models have been instrumental in characterizing the optical parameters and investigating the mechanisms behind induced myopia. Hypotheses regarding myopic development have been based on studies of the chick eye, although the absence of a fovea (a central area of high visual acuity) and retinal blood supply in chicks raises questions about the applicability of these findings to other species. Even closely related primate species can exhibit varying responses to conditions that induce form deprivation, suggesting differences in the mechanisms controlling eye growth. For example, monocular occlusion in the rhesus macaque leads to myopia when the ciliary muscle is paralyzed or the optic nerve is cut, but this does not occur in the stump-tailed macaque. These differences imply that excessive accommodation plays a role in myopia development in the stump-tailed macaque but not in the rhesus macaque.
Considering the variability among animal models, it is crucial to continually evaluate the relevance of each model to the human condition in ongoing myopia research. One promising avenue for future investigation lies in studying changes in gene ex-pression patterns during emmetropization and myopic progression, both within and across species. Identifying the deficient steps in the regulatory pathway would be a significant breakthrough given the substantial impact of myopia”.
- Summers JA, Schaeffel F, Marcos S, Wu H, Tkatchenko AV. Functional integration of eye tissues and refractive eye development: Mechanisms and pathways. Exp Eye Res. 2021 Aug;209:108693. doi: 10.1016/j.exer.2021.
- Schaeffel F, Feldkaemper M. Animal models in myopia research. Clin Exp Optom. 2015 Nov;98(6):507-17. doi: 10.1111/cxo.12312.
- 6.
- 7.
- 8.
- Vurgese, S., Panda-Jonas, S., & Jonas, J. B. , Scleral mechanisms of ocular axial elongation in myopia development and experimental treatment. Survey of Ophthalmology 2018, 63(6), 677-697.
- Pugazhendhi S, Ambati B, Hunter AA. , Pathogenesis and Prevention of Worsening Axial Elongation in Pathological Myopia. Clin Ophthalmol 2020, 18 (14), 853-873.
- 11.
- 12.
- 13.
- 14.
- 15.
- 12. 16.
- 13. 17
- 14. 18.
- 15. 19.
- 16. 20.
- 17. 21.
- 18. 22.
19-26. 23-30.
27-30. 31-34.
- 31. 35.
- 32. 36.
33-35. 37-39.
- 36. 40.
- 37. 41.
- 38. 42.
39-41. 43-45.
- 42. 46.
43-45. 47-49.
- 46. 50.
47, 48. 51, 52.
- 49. 53.
50, 51. 54, 55.
52-55. 56-59.
- 56. 60.
- 57. 61.
- 58. 62.
- 59. 63.
- We acknowledge your comment regarding the need for additional details on the experimental setup and animal model used in our study. To address this, we have included more information regarding the characteristics of TGM-2 mutant mice, such as their genetic background and also provided details on the age and sex distribution of the animals used. We have already described the procedure for inducing negative lens-induced myopia in mice under Section 2.2: 2.2. Murine Myopia Model.
Method Section 2.1; Pg 123-127: “We used 6 pairs of breeding mice in each strain to generate more knockout animals. These animals were systemically healthy and generate offspring readily. A background breeding pair was available for backcrossing in every 3-4 months to maintain the homozygosity. Both male and female mice at age of post-natal day 10 was utilized to induce experimental myopia and experiments end at age of post-natal day 52”.
- Thank you for suggesting the inclusion of more information about the muscarinic antagonists used in the in-vivo drug treatment and their mechanisms of action. We have now expanded our discussion in the Results section to provide a better context for the results obtained. This includes additional details on the muscarinic antagonists, their specific mechanisms of action, and their relevance to the study findings. These additions will contribute to a more comprehensive understanding of our research outcomes.
Included under Results Section 3.3; Pg 379-404: “Atropine sulfate is a competitive antagonist of the muscarinic acetylcholine receptors (M1-M5) found in various tissues of the body. It blocks the binding of acetylcholine to these receptors, thereby inhibiting the parasympathetic (cholinergic) nerve impulses. It produces a broad range of pharmacological effects, including reducing salivary and respiratory secretions, dilating pupils and relaxing smooth muscles.
Himbacine is a muscarinic antagonist that selectively targets the M2 and M3 subtypes of muscarinic receptors. By blocking these receptors, himbacine inhibits the effects of acetylcholine, which is the primary neurotransmitter of the parasympathetic nervous system. This blockade results in the relaxation of smooth muscles, reduction in glandular secretions, and other pharmacological effects similar to those of other muscarinic antagonists.
AFDX-116 is a selective antagonist of the M2 and M3 subtypes of muscarinic receptors. It blocks the binding of acetylcholine to these receptors, preventing the activation of downstream signalling pathways. This blockade leads to a reduction in parasympathetic activity, resulting in the relaxation of smooth muscles, inhibition of glandular secretions, and other effects associated with muscarinic antagonism.
Darifenacin is a selective antagonist of the M3 subtype of muscarinic receptors, which are primarily found in the bladder smooth muscle. By blocking these receptors, darifenacin reduces the bladder's contractions, thereby increasing its capacity and reducing the frequency of urination. It is primarily used to treat overactive bladder symptoms.
4-DAMP exhibits high selectivity for the M3 receptor and M5 receptor subtypes. These receptors play essential roles in various physiological processes, such as smooth muscle contraction, glandular secretion, and autonomic nervous system regulation. By antagonizing these receptors, 4-DAMP can exert specific effects on these systems.
Its mechanism of action involves blocking the binding of ACh to the receptor, inhibiting G-protein coupling, and thereby antagonizing the receptor-mediated physiological responses associated with these receptor subtypes.
Carbachol is a direct-acting cholinergic agonist that mimics the effects of acetylcholine. Although it is not a muscarinic antagonist, it is mentioned here for reference and used as a positive control. Carbachol stimulates muscarinic receptors, leading to the activation of parasympathetic responses. It is used in ophthalmology to constrict the pupils during certain procedures and in the treatment of glaucoma by reducing intraocular pressure”.
Pg: 416-418: “These findings suggest that TGM-2 mediates the effects of mAChR2/3 antagonist in this myopia model to reduce myopia”.
- Regarding the ordering of the Results section, we appreciate your feedback on the logical flow of presenting the findings. We have carefully restructured the Results section to ensure a more coherent and cohesive presentation of our results. By organizing the results based on the research question and the hypothesis being tested, we believe that the revised ordering will enhance the clarity and readability of the manuscript.
Agree on the suggestion flow and revised Figure 1 as 2 and 2 as 1 in this revision. Findings relating to the TGM-2 gene and the development of experimental myopia are presented as Figure 1 and moved original Figure 1 as Figure 2 and subsequent experiments on gene expression, proteomics and drug treatment.
Lastly, we are grateful for your confirmation that appropriate statistical tests were employed in our study. We have ensured that the statistical analyses and tests used, including Student's t-test, Mann-Whitney U test, and ANOVA, are correctly presented and ordered according to the context provided.
Once again, we sincerely appreciate your time, effort, and valuable feedback in reviewing our manuscript. We believe that the revisions we have made in response to your suggestions have significantly improved the overall quality and comprehensibility of the manuscript. We hope that the revised version will meet your expectations and align with the standards of the journal.

Reviewer 2 Report
The conclusion of the importance and significance of Atropine in treating Myopia is in my opinion controversial. One should explain it in more details.
I would have expected to see if there were any signs of side effects on the treated mice. This issue is significant if one wants to advance in further studies to treating humans.
Please address these issues.
Author Response
Dear Reviewer,
Thank you for taking the time to review our research on “Molecular Basis of Transglutaminase-2 and Muscarinic Cholinergic Receptors in Experimental Myopia: Target for Myopia Treatment”. We appreciate your valuable feedback and constructive comments. In response to your concerns, we would like to provide further clarification and address the issues raised.
Regarding the conclusion on the importance and significance of Atropine in treating Myopia, we understand that you find it controversial. We apologize if our explanation was not sufficiently detailed to support our viewpoint. In our revised manuscript, we will provide a more thorough explanation of the reasons behind our conclusion, taking into account the existing body of literature and the specific findings from our study.
Atropine (pan muscarinic blocker), a medication traditionally used to dilate the pupils and relax the focusing muscles of the eye, has been found to slow down the progression of myopia in children.
Several recent studies have explored the effectiveness of atropine in managing myopia progression. Here are some important findings:
Slows down myopia progression: Research has consistently shown that low-dose atropine eye drops can significantly slow down the progression of myopia in children. The Atropine for the Treatment of Myopia (ATOM) studies conducted in various countries, such as Singapore, Taiwan, and the United States, have demonstrated the efficacy of atropine in reducing myopia progression by approximately 50-60%.
Long-term effectiveness: Atropine has shown promising long-term effects in controlling myopia. Studies have reported that the beneficial effects of atropine persist even after the discontinuation of treatment. However, the optimal duration and dosage of atropine therapy are still under investigation.
Safety profile: Low-dose atropine has generally been found to be safe for children, with minimal systemic side effects. Some temporary ocular side effects, such as near vision blur and light sensitivity, may occur but are typically well-tolerated. Higher concentrations of atropine may result in more side effects, such as accommodation disturbances and photophobia.
Mechanism of action: The exact mechanism by which atropine slows down myopia progression is not fully understood. However, it is believed that atropine acts by inhibiting the growth of the eye, specifically the elongation of the eyeball that leads to myopia. Atropine may also modulate the balance of neurotransmitters in the retina, which could contribute to its anti-myopia effects.
Personalized treatment approach: Recent research has focused on identifying factors that can help personalize atropine treatment. Factors such as age, baseline myopia, and genetic factors have been studied to determine the optimal dose and duration of atropine therapy for individual patients. This personalized approach aims to maximize the benefits while minimizing potential side effects.
It is important to note that research on atropine for myopia treatment is still ongoing, and new findings may have emerged. In our study, we aim to identify TGM-2 as a new drug target for myopia treatment.
Included in this revision under conclusion section; Pg 745-761: “Several recent studies have explored the effectiveness of atropine in managing myopia progression. Although the involvement of muscarinic receptor subtypes in scleral remodelling is recognized, the precise downstream molecules remain unknown. Our research has revealed that transglutaminase (TGM)-2, a molecule previously associated with wound healing and cell migration, is up-regulated in the myopic murine sclera compared to the control. Additionally, we found that the gene expression of TGM-2 is down-regulated by atropine, an anti-muscarinic drug. TGM-2 has also been implicated in regulating extracellular matrix molecules such as fibronectin and collagen. Thus, we propose to investigate the central role of TGM-2 in the development of experimental myopia in mice by inducing myopia in TGM-2 deleted mice and mice treated with TGM inhibitors. We obtained supporting data in the form of TGM-2 associated transcripts and proteins in the sclera, as well as growth-related changes of these molecules in TGM-2 deleted mice. Furthermore, we examined the role of TGM-2 in muscarinic receptor signalling by using TGM inhibitors in mice with deleted muscarinic receptor subtypes prior to inducing myopia. If TGM-2 or its associated molecules are found to contribute to scleral elongation in experimental myopia, they could serve as the basis for novel pharmacological interventions for human myopia.
Pg 774-781: Our goal is to reduce the side effects associated with atropine, the current standard of care for school myopes, by elucidating the molecular mechanism of Transglutaminase-2 and Muscarinic Cholinergic Receptors in Experimental Myopia. This understanding will enable us to develop new drugs, either as monotherapies involving low doses of TGM-2 inhibitors or as combination therapies comprising low doses of TGM-2 inhibitors and atropine, to minimize potential side effects. Prior to translating these interventions to human trials in the future, we will conduct short- and long-term studies in preclinical models to assess safety and efficacy”.
Furthermore, you raised a valid point regarding the lack of information regarding potential side effects on the treated mice. We acknowledge the importance of assessing side effects as an integral part of advancing research from animal models to human trials. In our study, we utilized more than 100 mice and we didn’t observe any systemic or local toxicity in this short-term study. We will also emphasize the need for further investigation with long-term drug study into the safety profile of Atropine and TGM-2 compounds in order to ensure its viability as a treatment option for humans.
Thank you once again for your valuable input. We believe that addressing these concerns will strengthen the overall quality and impact of our research.
